# Objective stress values during radiation emergency medicine for future human resources: Findings from a survey of nurses

**Keita Iyama** [1,2]*, **Yoshinobu Sato**[2], **Takashi Ohba**[2,3], **Arifumi Hasegawa**[1,2]

**1** Department of Radiation Disaster Medicine, Fukushima Medical University, Fukushima, Japan, **2** Disaster Medicine Department, Fukushima Medical University Hospital, Fukushima, Japan, **3** Department of Radiation Health Management, Fukushima Medical University, Fukushima, Japan

* iyama-ngs@umin.net

## Abstract

### Introduction

Shortages of human resources in radiation emergency medicine (REM) caused by the anxiety and stress of due to working in REM, are a major concern. The present study aimed to quantify stress and identify which tasks involved in REM response are most stressful to help educate (human resource development) and effectively reduce stress in workers. Furthermore, the final goal was to reduce the anxiety and stress of medical personnel in the future, which will lead to sufficient human resources in the field of REM.

### Methods

In total, 74 nurses who attended an REM seminar were asked to answer a questionnaire (subjective) survey and wear a shirt-type electrocardiogram (objective survey). Then, informed consent was obtained from 39 patients included in the analysis. In the objective survey, average stress values of participants for each activity during the seminar were calculated based on heart rate variability (HRV). The average stress value was output as stress on a relative scale of 0–100, based on the model which is the percentile of the low-frequency/high-frequency ratio derived from HRV at any point in time obtained over time.

### Results

A total of 35 (89.7%) participants answered that they had little or no knowledge of nuclear disaster and 33 (84.6%) had more than moderate anxiety. Stress values observed during the decontamination process were significantly higher than those observed when wearing and removing protective gear and during the general medical treatment process ($P = 0.001$, 0.004, and 0.023, respectively). Stress values did not increase during general medical treatment performed in protective clothing, but increased during the decontamination process, which is the task characteristic of REM.

**Data Availability Statement:** Our data may be personally identifiable and we have not been approved by the Ethics Committee to freely share our data. The contact information for ethics

committee is as follows. Fukushima Medical University Ethics Committee e-mail: rs@fmu.ac.jp Tel: +81-24-547-1825 Fax: +81-24-581-5163.

**Funding:** KI received the grant by JSPS KAKENHI Grant-in-Aid for Early-Career Scientists (20K17870) (https://www.jsps.go.jp/j-grantsinaid/), and Radiation Disaster and Medical Science Research Center, FY2021 (KY2021220) (https://housai.hiroshima-u.ac.jp). AH received the grant by, JSPS KAKENHI Grant-in-Aid for Scientific Research(B) (19H03762) (https://www.jsps.go.jp/j-grantsinaid/). The funders had no role in study design, data collection and analysis, decision to publish, or preparation of the manuscript.

**Competing interests:** The authors have declared that no competing interests exist.

## Discussion

Stress felt by medical personnel throughout the entire REM response may be effectively reduced by providing careful education/training to reduce stress during the decontamination process. Reducing stress during REM response effectively could contribute to resolving the shortage of human resources in this field.

## 1. Introduction

After the 1990s, the risk of encountering different hazardous events has been increasing worldwide [1–3]. A rapid response is essential to save the lives of injured or critical patients in times of disaster or emergency situations [4–6] and requires expansion of the surge capacity, which is the ability to respond to emergencies [7–10]. The most important factor in expanding the surge capacity is securing manpower (human resources) [7–10]. However, the intention of disaster responders to engage in specific disasters including chemical, biological, radiological, nuclear, and explosive hazards, such as the coronavirus disease 2019 pandemic, chemical terrorism, and nuclear disaster, is low compared with that of natural disasters, such as earthquakes and floods [11–15]. Such low-level intention to engage is a major cause of human resource shortages during specific disaster responses and is the biggest obstacle to response activities for specific disasters. In particular, nurses account for approximately 40% of healthcare workers, and understanding their current status is crucial for efficiently securing human resources for the entire healthcare workforce [16].

The nuclear disaster caused by an accident at the Fukushima Daiichi Nuclear Power Plant (FDNPP) after the Great East Japan Earthquake on March 11, 2011, is one of the most disastrous events in recent years. Crisis response professionals have concerns about the risk of radiation exposure in the general population after the FDNPP accident. During the initial response, hospital staff at medical institutions receiving victims experienced strong feelings of anxiety due to the risk of radiation exposure, which affected the smooth acceptance of patients in the hospital [17]. It was shown that the intention to engage in nuclear disaster medicine decreases as a result of anxiety due to radiation being an invisible hazard, as well as a lack of knowledge and skills [14]. Nurses were previously reported to be highly anxious about radiation due to a lack of knowledge about radiation and medical exposure [18–20]. Therefore, the shortage of human resources in this field is attributed to the negative chain of events caused by anxiety and stress (Fig 1). Lack of knowledge causes radiation exposure-related anxiety among workers, which reduces their intention to engage in radiation emergency medicine (REM) activities [14]. This resulted in a shortage of human resources and educators, thereby making it impossible to provide sufficient education and increase the knowledge of workers [14, 21–24]. Hence, this negative chain should be suppressed to smoothly provide nuclear disaster medical services. After the FDNPP accident, several surveys and studies were conducted on anxiety and stress, which are important elements of this negative chain, targeting various populations, and previous subjective evaluation surveys using questionnaires were selected as the study method [25]. However, to date, no data has objectively presented a stress assessment of medical personnel engaged in the initial response to critical patients using physiological indicators.

Since the FDNPP accident, Fukushima Medical University (FMU) has been working to accept patients exposed to or contaminated with radiation related to the FDNPP accident. Furthermore, FMU continues to develop human resources related to REM and presents over 20 REM-related seminars annually [26, 27]. Moreover, FMU has also been developing human

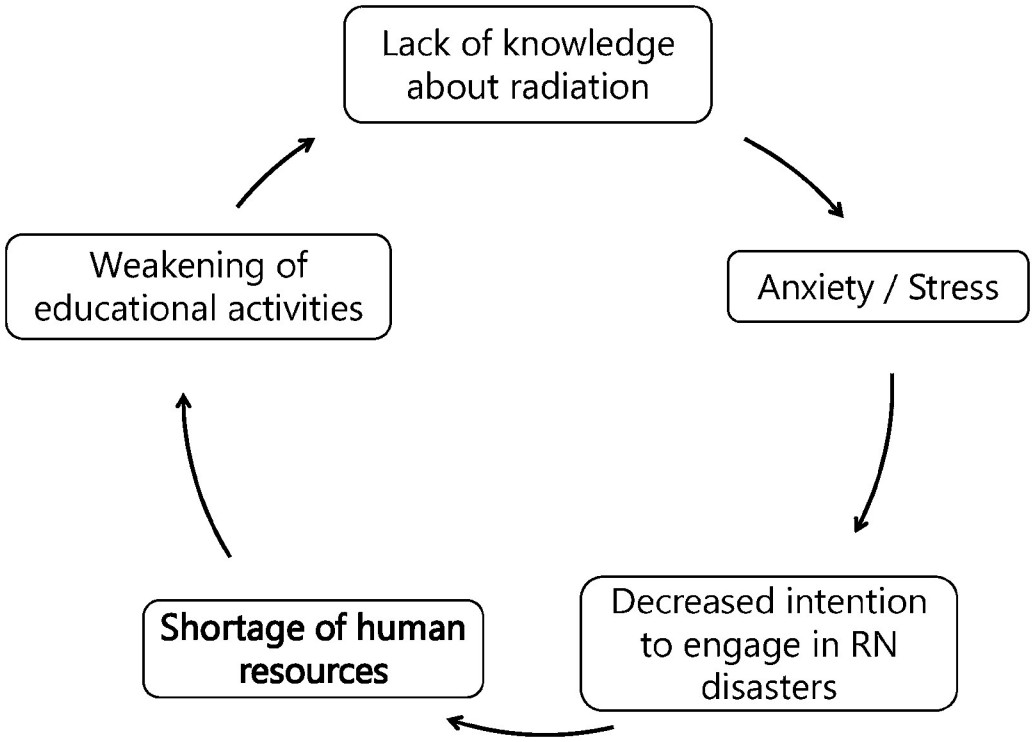

**Fig 1. Negative chain of anxiety and stress in radiation emergency medicine (REM).**

resources in the field of REM by establishing a joint master's program with Nagasaki University (the Joint Graduate School, Disaster and Radiation Medical Sciences Joint Master's Degree) [28]. The following important points have been identified: (1) medical personnel are anxious and stressed about engaging in REM due to a lack of knowledge and skills; (2) it is important to effectively reduce the anxiety and stress in order to resolve stress and anxiety; and (3) reducing stress and anxiety will contribute to the stable supply of human resources in the future and lead to smooth REM activities. Therefore, this study reported that objective assessment was the first measurement that could measure personnel stress using heart rate variability (HRV) during an REM seminar in FMU in addition to the traditional subjective assessment of anxiety and stress.

A widely known method for objectively indicating stress levels is the evaluation of sympathetic nerve activity based on HRV [29, 30]. HRV refers to the variation in the RR interval of the electrocardiogram, which indicates the instantaneous heart rate, and is affected by the balance between the sympathetic and parasympathetic nervous systems. The low-frequency/high-frequency ratio (LF/HF) obtained from the analysis of the HRV frequency is used as a surrogate parameter for sympathetic overactivity and is associated with stress load [31–33]. There are several reports on the validity of HRV in assessing stress levels [34–37]. Furthermore, with advances in technology, it has been suggested that wearable devices capable of measuring HRV can be used to determine stress levels [30, 38].

The purpose of the present study is to identify the points that need to be taken into consideration for maintaining sufficient human resources to achieve the final goal of facilitating nuclear disaster response activities. Previous reports have shown that education and training far beyond the current level can help achieve the abovementioned goal [39]. Therefore, to efficiently conduct education and training in the future, this study first objectively evaluated

actual stress status. The current study derived stress values from HRV data of nurses who attended the REM seminar of FMU. The objective stress values obtained from the HRV of the nurses were compared with the content of the practical training activities to identify the activities with the highest stress load. The stress values derived from the physiological indices of the medical personnel who attended the REM seminar are the best data to objectively indicate the knowledge and skills that should be focused on for effective personnel training to break the negative chain (Fig 1). Our data may serve as the basis for objective anxiety and stress indicators for responders in hazards that cannot be visualized, such as radiation or viruses.

## 2. Materials and methods

Three important points were extracted from past educational activities in the field of nuclear disaster. In order to achieve the final goal of facilitating nuclear disaster response activities and securing sufficient human resources, this study examined objective indicators using the following methods.

### 2.1. Study design

The REM seminar of FMU was held in Fukushima City, Japan, and it included individuals who might be engaged in radiation emergency medicine in case of an actual nuclear disaster. Hence, these participants were included in the study. The objective stress value of the participants was evaluated, and the values between each of the processes were compared to validate which process is the most stressful during the REM activity.

### 2.2. Participant selection

In total, 74 nurses who attended the REM seminar of FMU between October 2020 and February 2021 were asked to participate in this study. Among them, 39 who agreed to wear a shirt-type electrocardiogram and participate in the study were included in the subsequent analyses. The study was conducted after study details including the results and use of images were explained to the participants and informed consent was obtained.

### 2.3. Data collection

**2.3.1. Measurement of HRV over time and calculation of average stress values.** Wearable electrocardiogram devices can be used to quantitatively assess the stress of firefighters engaged in firefighting and rescue activities, which are physically and mentally demanding [40]. In the present study, a shirt-type electrocardiogram (Mitsufuji Corp. Tokyo, Japan. https://www.mitsufuji.co.jp/service/) was used to collect HRV data from the study participants, and stress values were derived from the LF/HF obtained from the frequency analysis of HRV using Hamon application software (Mitsufuji Corp. Tokyo, Japan). Using the individual participants' LF/HF data obtained during the period of measurement, a normal distribution model of each participants' LF/HF values was constructed. Based on the model, the percentile of the LF/HF value at any point in time obtained over time was output in the range of 0 to 100, which was the stress value used in this study.

The content of the REM seminar of FMU is listed below in chronological order: lecture (15 min); dressing in protective gear (10 min); patient acceptance (3 min); general medical treatment (10 min); decontamination (12 min); and undressing from protective gear (6 min) (Fig 2). The decontamination process in the REM seminar of FMU includes undressing patients with injury, dry decontamination, which involves wiping the body surface, and wet decontamination, which comprises wound cleaning. Similar to a previous report, the trainees performed

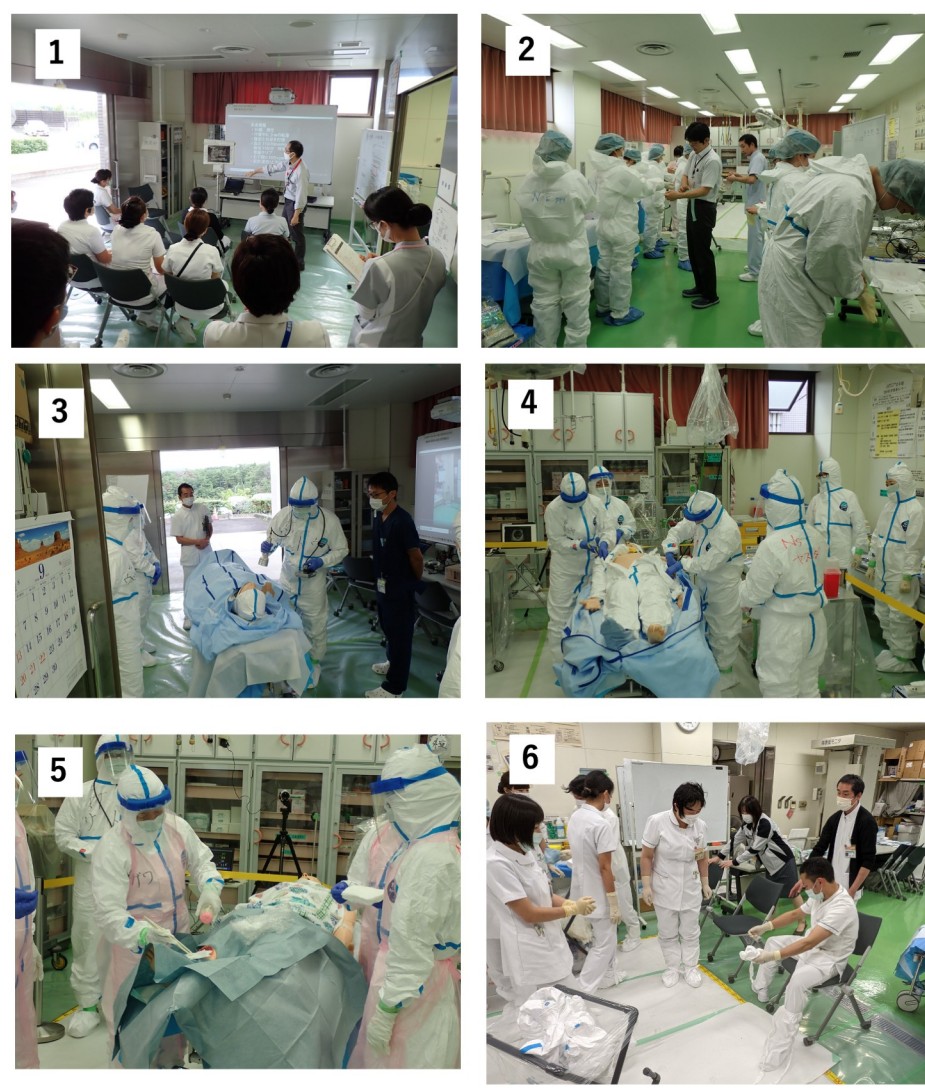

**Fig 2. Seminar image of each process.** (1) Lecture, (2) dressing in protective gear, (3) patient acceptance, (4) general medical treatment, (5) decontamination, and (6) undressing from protective gear.

decontamination, which prevented the spread of contamination [41]. Study participants wore a shirt-type electrocardiogram during the seminar and the average stress value for each activity during the seminar was automatically calculated based on HRV using Hamon application software. This was an observational cross-sectional study that compared differences in stress values between each of the abovementioned processes.

**2.3.2. Questionnaire survey on the background characteristics of the participants.** Questionnaires were administered and filled out by participants before and after the seminar on the same day. Yamada et al. [42] reported that interest in emergency medicine and experience in disaster activities are related to an interest in nuclear disaster medicine, and the questionnaire in our study was developed with this in mind. The self-administered questionnaires were placed into collection boxes. The following items were assessed: age, sex, marital status (yes/no), with children (yes/no), experience in emergency medicine (4-point Likert scale), experience in disaster response activities (yes/no), interest in emergency medicine (yes/no),

interest in nuclear disaster medicine (yes/no), recognition of nuclear disaster (4-point Likert scale), presence of acceptable radiation dose to oneself (yes/no), and level of anxiety caused by working during a nuclear disaster event (4-point Likert scale). Participants were required to respond to the questions anonymously to prevent influencing responses.

### 2.4. Statistical analysis

The background factors of the participants were compared between the groups with and without an interest in nuclear disasters using *t* test for age and Fischer's exact test for other background factors. Stress values were compared using *t* test and Tukey–Kramer test for multiple comparisons. All statistical analyses were performed using JMP pro 16 (SAS Institute Inc., Cary, NC, USA) with the significance level set at 0.05.

### 2.5. Ethical considerations

The present study was conducted in accordance with the Declaration of Helsinki and was approved by the Ethics Committee of FMU (approval number: 2019–290). The individual in this manuscript has given written informed consent to publish.

## 3. Results

The background of the study participants is shown in Table 1. Most participants were female and there were no significant differences in background factors between the groups with and without an interest in nuclear disasters, other than interest in emergency medicine ($P < 0.001$) and recognition of nuclear disaster ($P = 0.008$).

The mean stress values during each activity are shown in Fig 3 and Table 2. The stress value during the decontamination process was significantly higher than that when wearing the protective gear and removing it and during the general medical treatment process ($P = 0.001$, $P = 0.004$ and $P = 0.023$, respectively; Table 2). No significant increase in mean stress value was observed during general medical treatment, even with wearing protective gear, and stress values only increased during tasks specific to REM. The results of the comparison of mean stress values between the groups with and without interest in nuclear disaster are shown in Table 3. There were no significant differences in the stress values for each process between the two groups. The assessment of anxiety levels at the time of participating in nuclear disaster response activities revealed that the number of participants who answered that they were anxious after the seminar (n = 20) was significantly lower than that before the seminar (n = 33; $P = 0.010$) (Table 4).

## 4. Discussion

The present study used physiological indices (HRV) to objectively quantify stress in medical personnel who provide REM care, which has been previously surveyed using subjective questionnaires. We objectively clarified which behaviors caused stress during REM work, with an aim to efficiently reduce stress to secure human resources to break the negative chain of anxiety and stress (Fig 1). Stress values differ depending on activities and some specific processes should be focused on to effectively reduce the stress experienced by medical personnel in future. The derived stress values in the present study revealed that the decontamination process imposes the highest stress load on those involved in REM (57.9) (Table 2, Fig 3). We also revealed that interest in nuclear disasters did not significantly affect the stress value ($P = 0.270–0.617$) (Table 3). In other words, stress values increased during the

**Table 1. Characteristics of participants.**

| | Interested in nuclear disaster medicine | Not interested in nuclear disaster medicine | P-value |
|---|---|---|---|
| | n = 17 | n = 22 | |
| Age (years), mean (SD) | 41.6 (8.4) | 41.1 (8.4) | 0.865 |
| **Sex, n (%)** | | | 1.000 |
| Female | 21 (95.5) | 16 (94.1) | |
| Male | 1 (4.5) | 1 (5.9) | |
| **Married, n (%)** | | | 0.106 |
| Yes | 12 (70.6) | 9 (40.9) | |
| No | 5 (29.4) | 13 (59.1) | |
| **With a child, n (%)** | | | 0.343 |
| Yes | 12 (70.6) | 12 (54.5) | |
| No | 5 (29.4) | 10 (45.5) | |
| **Experience in emergency medicine, n (%)** | | | 0.449 |
| Currently engaged | 3 (17.7) | 1 (4.6) | |
| Previously engaged | 3 (17.7) | 4 (18.2) | |
| Not much engaged | 4 (23.5) | 3 (13.6) | |
| Not at all | 7 (41.2) | 14 (63.6) | |
| **Experience in disaster response activities, n (%)** | | | 0.074 |
| Yes | 3 (17.7) | 0 (0.0) | |
| No | 14 (82.3) | 22 (100) | |
| **Interest in emergency medicine, n (%)** | | | **<0.001** |
| Yes | 16 (94.1) | 2 (9.1) | |
| No | 1 (5.9) | 20 (90.9) | |
| **Recognition/knowledge of nuclear disaster, n (%)[†]** | | | **0.008** |
| Well aware | 0 (0.0) | 0 (0.0) | |
| Somewhat aware | 4 (23.5) | 0 (0.0) | |
| Minimally aware | 13 (76.5) | 19 (86.4) | |
| Not aware | 0 (0.0) | 3 (13.6) | |
| **Anxiety during nuclear disaster response activities, n (%)[†]** | | | 0.585 |
| Extreme | 1 (5.9) | 3 (13.6) | |
| Moderate | 12 (70.6) | 17 (77.3) | |
| Minimal | 3 (17.7) | 2 (9.1) | |
| None | 1 (5.9) | 0 (0.0) | |
| **Established own acceptable standards of radiation exposure dose, n (%)** | | | 1.000 |
| Yes | 1 (5.9) | 1 (4.6) | |
| No | 16 (94.1) | 21 (95.4) | |

P-values in bold and italics are statistically significant.

[†]; These responses were assessed using a 4-point Likert scale.

decontamination process regardless of original interest, suggesting that focused education/training on the decontamination process is necessary to effectively reduce stress during REM treatment for all.

Although the present study aimed to reduce anxiety and stress, it is important to note that a certain amount of anxiety stress contributes to the safety of disaster responders. In particular, the mean stress value during wearing and removing protective gear were the lowest among the

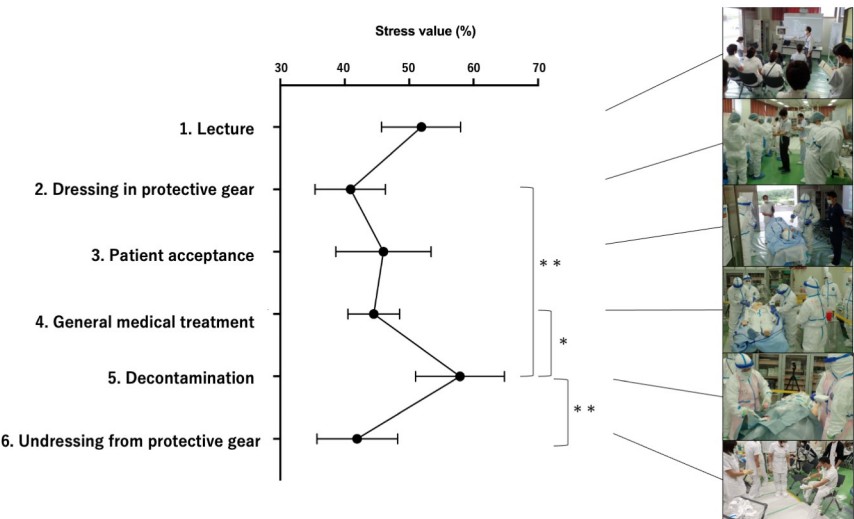

**Fig 3. Average stress values of each process.** Error bars indicate 95% confidence intervals. *; $P < 0.05$, **; $P < 0.01$.

six items (40.9 and 41.9, respectively) (Table 2). This is likely due to the reduced attention to properly wearing protective gear or the relief of having finished dealing with a contaminated patient. However, this sense of relief may lead to a careless undressing from their own protective gear, and in that situation, may lead to self-contamination. Although it is not clear whether this occurs during actual REM treatment, the results of the present study support previous reports of increased risk of contamination during undressing [43–45]. It should be noted that moderate anxiety and stress are not necessarily harmful, and it is important to emphasize that REM includes undressing and restoration of the treatment space in education and training.

Previous studies have shown that nurses with more knowledge about radiation tend to have better mental health in terms of anxiety about radiation exposure, and that education/acquisition of knowledge about the health risks of radiation exposure is important for medical personnel [46]. Furthermore, a previous study has reported that interest influences knowledge [47]. However, the present study found no difference in stress values between the group interested in nuclear disasters and that with no interest ($P = 0.270–0.617$) (Table 3). This result suggests that knowledge is affected by interest but stress is not affected directly. In addition, the interested group also showed increased stress values during the decontamination process, which objectively suggests that educational activities to reduce stress during the

**Table 2. Mean stress values of each process.**

| No. | Action | Stress value (%) | 95% CI | P-value | | | | |
|---|---|---|---|---|---|---|---|---|
| | | Mean (SD) | | vs. 2 | vs. 3 | vs. 4 | vs. 5 | vs. 6 |
| 1 | Lecture | 51.9 (18.7) | 45.7–58.0 | 0.0983 | 0.7303 | 0.4965 | 0.7210 | 0.1815 |
| 2 | Dressing in protective gear | 40.9 (16.3) | 35.4–46.3 | — | 0.8310 | 0.9533 | ***0.0013*** | 0.9999 |
| 3 | Patient acceptance | 46.0 (22.1) | 38.6–53.4 | — | — | 0.9993 | 0.0652 | 0.9322 |
| 4 | General medical treatment | 44.5 (12.1) | 40.5–48.5 | — | — | — | ***0.0232*** | 0.9901 |
| 5 | Decontamination | 57.9 (20.4) | 51.0–64.8 | — | — | — | — | ***0.0036*** |
| 6 | Undressing from protective gear | 41.9 (18.4) | 35.7–48.2 | — | — | — | — | — |

CI, confidence interval; SD, standard deviation. *P*-values in bold italics are statistically significant.

**Table 3. Comparison between the groups with and without interest in nuclear disaster.**

| Action | Stress value (%), mean (SD) | | P-value |
|---|---|---|---|
| | **Interested in nuclear disaster medicine** | **Not interested in nuclear disaster medicine** | |
| | **n = 17** | **n = 22** | |
| Lecture | 53.9 (18.7) | 50.1 (19.1) | 0.560 |
| Dressing in protective gear | 39.3 (12.6) | 42.1 (18.9) | 0.617 |
| Patient acceptance | 49.3 (25.5) | 43.5 (19.3) | 0.440 |
| General medical treatment | 47.1 (13.8) | 42.6 (10.7) | 0.270 |
| Decontamination | 60.7 (19.0) | 55.6 (21.6) | 0.465 |
| Undressing from protective gear | 38.4 (19.8) | 44.8 (14.2) | 0.303 |

SD, standard deviation.

decontamination process are the most efficient way to reduce stress throughout REM activities, regardless of the background of the medical personnel. Moreover, misunderstandings and a lack of formal education in the field of nuclear disaster can also increase anxiety and fear of radiation-related accidents. In a previous study that assessed the willingness of health care workers to come to work, radiation was found to be the risk factor that kept most workers at home [48]. In the field of radiation disaster medicine, the Dunning–Kruger effect, which is a cognitive bias known as the illusion of superiority and is associated with the relationship between knowledge and self-confidence, suggests that anxiety and stress are expected to be related to amount of knowledge and experience (Fig 4) [49]. In other words, the acquisition of moderate knowledge and experience may adversely contribute to radiation anxiety and stress. The participants in the present study were nurses working in general wards (not radiation-related departments), and 35 (89.7%) answered that they had little or no knowledge and experience of nuclear disasters (Table 1). Therefore, it is highly likely that the study participants had no knowledge and experience of nuclear disasters originally and even a short seminar, such as this one, could have provided knowledge and experience that lead to a significant reduction in their anxiety levels ($P = 0.010$) (Table 4).

As mentioned in the previous paragraph, it was assumed that the participants of this study would have no knowledge or experience of nuclear disasters, and the content of the REM seminar of FMU was very basic. Seminar contents vary greatly depending on the participants' basic knowledge and experience level. Various REM-related seminars are held worldwide for providing more advanced training, and some of the longer ones take 4 days [50]. Those who attended the advanced training tended to expect that the seminar would teach them about

**Table 4. Comparison of anxiety before and after attending the seminar.**

| | Before seminar | After seminar | P-value |
|---|---|---|---|
| | **n = 39** | **n = 39** | |
| **Anxiety during nuclear disaster response activities, n (%)[†]** | | | ***0.010*** |
| Extreme | 4 (10.3) | 2 (5.1) | |
| Moderate | 29 (74.4) | 18 (46.2) | |
| Minimal | 5 (12.8) | 17 (43.6) | |
| None | 1 (2.5) | 2 (5.1) | |

P-value in bold and italics is statistically significant.

[†]; These responses were assessed using a 4-point Likert scale.

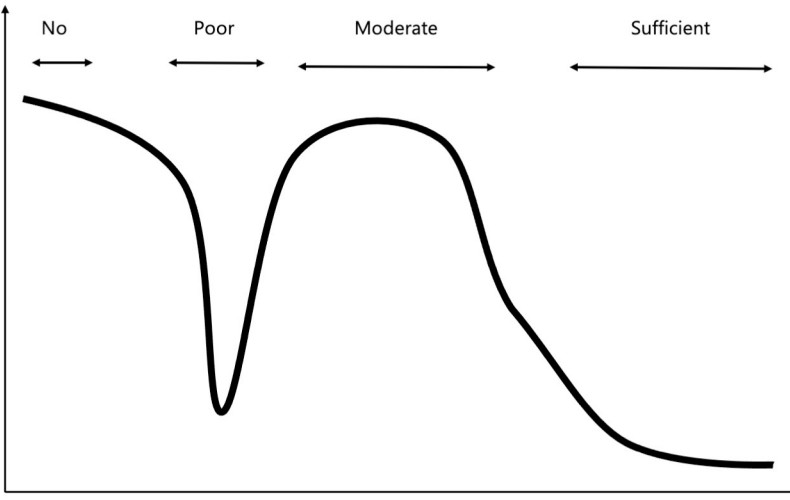

**Fig 4. Diagrammatic image of the association between knowledge and anxiety/stress in radiation emergency medicine.** People without knowledge/experience (no knowledge/experience) have high anxiety/stress levels, which decreases after gaining minimal knowledge/experience (poor knowledge/experience). As they gain more knowledge/experience (moderate knowledge/experience), their anxiety/stress levels increase. As they acquire more knowledge/experience (sufficient knowledge/experience), their anxiety/stress levels will decrease according to the amount of knowledge/experience.

regulation and medical manuals rather than procedures [41]. Therefore, it is desirable to develop a curriculum for each seminar that satisfies the needs of the target participants. Since it is considered ideal for participants to attend seminars with a moderate sense of tension, the results of this study suggest that the participants' HRV may be useful in evaluating curriculum content. HRV could also be used as an indicator of proficiency for those who repeat the training (i.e., for procedures regarding which the participant has gained knowledge in the past and those that the participant has experience performing and has the confidence to perform, with no increase in stress values as an indicator of proficiency). It is expected that studies will be conducted on specific educational methods to accurately educate medical personnel without causing excessive increase in anxiety/stress in the future.

## 4.1. Limitation

The present study evaluated stress values of seminar participants during a seminar, which may differ from the stress experienced by medical personnel in actual clinical situations. However, nuclear disaster events are rare and it is difficult to measure stress during actual disaster activities.

REM includes some processes such as dressing and undressing one's own protective gear and general patient care. However, based on the sequence of processes, the stress value was significantly high in the decontamination process. To efficiently reduce stress via REM trainings, it is necessary to decrease stress during the decontamination process. Further, education and training are essential in efficiently reducing stress during decontamination. However, specific methods for decreasing stress have not been validated. To reduce anxiety/stress, it might be important to inform participants during REM trainings that not only dosimetry with personal dosimeters but also dosimetry with chromosomes can be performed in case of radiation

exposure [51]. In the future, factors associated with stress during the decontamination process should be identified to facilitate concrete measures. Nevertheless, further studies must be conducted to identify factors affecting stress levels to determine specific strategies for reducing stress effectively.

### 4.2. Conclusion

The use of a shirt-type electrocardiogram made it possible to assess stress levels during the activities presented at the REM seminar. The stress load was highest during the decontamination process. It is important to effectively reduce stress during the decontamination process in future to break the negative chain of anxiety and stress via REM (Fig 1) and resolve the shortage of human resources. Since factors that affect stress during decontamination process have not yet been clarified, it is desirable to clarify these factors in future so that concrete measures to efficiently reduce stress during decontamination process can be clarified, and eventually the shortage of human resources can be resolved and nuclear disaster activities can be performed smoothly.

### Acknowledgments

The authors are extremely grateful to the study participants and all the staff involved in conducting the seminar. Additionally, the authors would like to thank Mr. T. Abe and Mr. T. Matsumoto for their advice and help with the analysis.

### Author Contributions

**Conceptualization:** Keita Iyama, Takashi Ohba, Arifumi Hasegawa.

**Data curation:** Keita Iyama, Yoshinobu Sato.

**Formal analysis:** Keita Iyama, Yoshinobu Sato.

**Funding acquisition:** Keita Iyama, Arifumi Hasegawa.

**Methodology:** Keita Iyama.

**Project administration:** Keita Iyama, Yoshinobu Sato.

**Resources:** Keita Iyama, Yoshinobu Sato, Takashi Ohba.

**Supervision:** Arifumi Hasegawa.

**Visualization:** Keita Iyama.

**Writing – original draft:** Keita Iyama.

**Writing – review & editing:** Keita Iyama, Yoshinobu Sato, Takashi Ohba, Arifumi Hasegawa.

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
