## [Decision Letter · Decision Letter 0]

10 May 2022

PONE-D-22-11048Objective stress values during radiation emergency medicine for future human resourcesPLOS ONE

Dear Dr. Keita Iyama

Thank you for submitting your manuscript to PLOS ONE. After careful consideration, we feel that it has merit but does not fully meet PLOS ONE’s publication criteria as it currently stands. Therefore, we invite you to submit a revised version of the manuscript that addresses the points raised during the review process.

We look forward to receiving your revised manuscript.

Kind regards,

Mohamad Syazwan Mohd Sanusi

Academic Editor

PLOS ONE

Journal Requirements:

3. We note that Figures 2 and 3 includes an image of a participant in the study. 

Reviewers' comments:

Reviewer's Responses to Questions

**Comments to the Author**

1. Is the manuscript technically sound, and do the data support the conclusions?

Reviewer #1: Yes

Reviewer #2: Yes

Reviewer #3: Partly

Reviewer #4: Yes

Reviewer #5: Yes

2. Has the statistical analysis been performed appropriately and rigorously? 

Reviewer #1: Yes

Reviewer #2: Yes

Reviewer #3: Yes

Reviewer #4: Yes

Reviewer #5: Yes

3. Have the authors made all data underlying the findings in their manuscript fully available?

Reviewer #1: Yes

Reviewer #2: Yes

Reviewer #3: Yes

Reviewer #4: Yes

Reviewer #5: Yes

4. Is the manuscript presented in an intelligible fashion and written in standard English?

Reviewer #1: Yes

Reviewer #2: No

Reviewer #3: Yes

Reviewer #4: Yes

Reviewer #5: Yes

5. Review Comments to the Author

Reviewer #1: I would like to thank authors for their interesting and timely useful study. It gain more practical knowledge as well as opportunity to intervention for keeping medical emergency treatment process intact.

Minor comment :

1.It would be better to address decontamination process in detail. This will further highlighted gaps and intervention methods.

2. Authors can recommend some of the strategies to overcome stress levels during the decontamination process in the discussion section.

Reviewer #2: The authors need to state the limitations of their study and recommendations for future studies. Further, a distinct section for conclusion is recommended. Furthermore, all tables need formatting and should be assessed for readability.

Reviewer #3: Queries Suggestions to the author

Methodology Method

Study design should be explain clearly :-

1. If Mixed method so, justify the reason behind such study designs for this article.

2. Kindly explain the justification for the sampling strategy used in the course of this study

3. Description of tool e.g questionnaire management should be mentioned clearly

The sample size is not matching…kindly check: line 29 & 120

Orderliness Kindly rearrange by following the scientific research process hence outlining the method used clearly.

Participants characteristics in the manuscript should be properly tabulated

Reviewer #4: The manuscript is clearly written and is scientifically sound. Recent literature published covering this topic of research include:

Carr Z, Maeda M, Oughton D, Weiss W. Non-radiological impact of a nuclear emergency: preparedness and response with the focus on health. Radiation protection dosimetry. 2018 Dec 1;182(1):112-9.

Kulka U, Wojcik A, Di Giorgio M, Wilkins R, Suto Y, Jang S, Quing-Jie L, Jiaxiang L, Ainsbury E, Woda C, Roy L. Biodosimetry and biodosimetry networks for managing radiation emergency. Radiation protection dosimetry. 2018 Dec 1;182(1):128-38.

There are a few recommendations that are suggested:

Introduction: The main aim of the study should be clearly highlighted in the last paragraph emphasising the significance of the study and what it adds to the existing knowledge base.

Methodology: The study design should be briefly explained and justified with a brief sub-heading

Discussion: The limitations of the study can be highlighted more clearly and its suggested that they are re-written.

Reviewer #5: Comments to the Author

I want to congratulate the authors on this clear and well-written manuscript.

Overall, this high-quality manuscript has implications for policymakers and other concerned bodies. But there are some points of consideration for revision, but these changes are not likely to be substantial. I, therefore, recommend the following minor revisions.

INTRODUCTION

- Well-written

- author state in line with 46 “In recent years” this is not a good word because what does it mean how many years are called as recent years, so please be specific or try to put the time period.

- the author needs to revise the sentence In line with “57- 63” the statement was to wordy please try to summarize it.

- the sentence in line with “80-84” have to written here in the introductory part, I think they are part of result of this study.

METHODS

- well-written

- there is a need of separate subheading which indicates study area, study design, participant selection, and data collection technique.

RESULT

- well-written

- How authors classified that the level of Anxiety as “Extreme, Moderate, Little, and None” this not clear, because there have to be predefined criteria or scale of measure to classify the anxiety level.

DISCUSSION

- Well-written

- LIMITATION OF THE STUDY: if possible, to mention it’s recommended.

- CONCLUSION AND RECOMMENDATION: - it’s good if there is conclusion and recommendation headings. Also, recommendation for future direction.

6. PLOS authors have the option to publish the peer review history of their article (what does this mean?). If published, this will include your full peer review and any attached files.

Reviewer #1: No

Reviewer #2: No

Reviewer #3: No

Reviewer #4: **Yes: **Dr Muslim Abbas Syed

Reviewer #5: **Yes: **Astawus Alemayehu

---

## [Author Response · Author response to Decision Letter 0]

6 Jun 2022

Reviewer #1: I would like to thank authors for their interesting and timely useful study. It gain more practical knowledge as well as opportunity to intervention for keeping medical emergency treatment process intact.

Minor comment :

1.It would be better to address decontamination process in detail. This will further highlighted gaps and intervention methods.

Thank you for the advice. The following sentences were added in the Material and Methods section:

“The decontamination process in the REM seminar of FMU includes undressing patients with injury, dry decontamination, which involves wiping the body surface, and wet decontamination, which comprises wound cleaning. Similar to a previous report, the trainees performed decontamination, which prevented the spread of contamination.”

2. Authors can recommend some of the strategies to overcome stress levels during the decontamination process in the discussion section.

Stress levels during the decontamination procedure must be reduced effectively via REM seminars. However, this study did not validate the specific method used to decrease stress levels during decontamination. Nevertheless, further research must be performed to validate factors affecting stress levels during decontamination procedures. The following sentences were added in the Limitation section:

“REM includes some processes such as dressing and undressing one’s own protective gear and general patient care. However, based on the sequence of processes, the stress value was significantly high in the decontamination process. To efficiently reduce stress via REM trainings, it is necessary to decrease stress during the decontamination process. Further, education and training are essential in efficiently reducing stress during decontamination. However, specific methods for decreasing stress have not been validated.”

Reviewer #2: The authors need to state the limitations of their study and recommendations for future studies. Further, a distinct section for conclusion is recommended. Furthermore, all tables need formatting and should be assessed for readability.

Thank you for the advice. We have added separate Limitation and Conclusion sections. Moreover, we have declared that this study is observation during just a seminar, not the real radiation emergency situation as study limitation. Further, it is unclear that what kind of factor affects the stress during the decontamination process. We also have added the separate section for conclusion. Moreover, the tables were modified.

Reviewer #3: Queries Suggestions to the author

Methodology Method

Study design should be explain clearly :-

1. If Mixed method so, justify the reason behind such study designs for this article.

Thank you for the valuable comments. This was a quantitative study comparing stress values between processes via the REM seminar. The following sentence was added in the Data collection section: 

“This was an observational cross-sectional study comparing differences in stress values between each of the abovementioned processes.”

2. Kindly explain the justification for the sampling strategy used in the course of this study

The following sentence was added in the Study design section:

The REM seminar of FMU was held in Fukushima City, Japan, and it included individuals who might be engaged in radiation emergency medicine in case of an actual nuclear disaster. Hence, these participants were included in the study.

3. Description of tool e.g questionnaire management should be mentioned clearly

Thank you for the advice. We have revised the sentences in the “2.2.2. Questionnaire survey on the background characteristics of the participants” section.

The sample size is not matching…kindly check: line 29 & 120

Thank you for pointing this out. The following sentences in the Abstract section were modified:

“In total, 74 nurses who attended the REM seminar were instructed to participate in a questionnaire (subjective) survey and to wear a shirt-type electrocardiogram (objective survey). Then, informed consent was obtained from 39 participants who were included in the analysis.”

Orderliness Kindly rearrange by following the scientific research process hence outlining the method used clearly.

Participants characteristics in the manuscript should be properly tabulated

The “2.1 Study design” section was added in the Methods section, and the Result section was rearranged (Table 1). Table 1 shows the characteristics of the participants.

Reviewer #4: The manuscript is clearly written and is scientifically sound. Recent literature published covering this topic of research include:

Carr Z, Maeda M, Oughton D, Weiss W. Non-radiological impact of a nuclear emergency: preparedness and response with the focus on health. Radiation protection dosimetry. 2018 Dec 1;182(1):112-9.

Kulka U, Wojcik A, Di Giorgio M, Wilkins R, Suto Y, Jang S, Quing-Jie L, Jiaxiang L, Ainsbury E, Woda C, Roy L. Biodosimetry and biodosimetry networks for managing radiation emergency. Radiation protection dosimetry. 2018 Dec 1;182(1):128-38.

Thank you for suggesting the abovementioned articles. Some data were added in the Discussion and Limitation sections, and their references were included. 

“The current study aimed to achieve the final goal of facilitating nuclear disaster response activities. Previous reports have shown that education and training far beyond the current level can help achieve the abovementioned goal. Therefore, to efficiently conduct education and training in the future, this study first objectively evaluated actual stress status. The results are applicable globally and can be utilized in REM training using the REMPAN network worldwide.”

“To reduce anxiety/stress, it might be important to inform participants during REM training that not only dosimetry with personal dosimeters but also dosimetry with chromosomes can be performed in case of radiation exposure.”

There are a few recommendations that are suggested:

Introduction: The main aim of the study should be clearly highlighted in the last paragraph emphasising the significance of the study and what it adds to the existing knowledge base.

Thank you for the advice.

The following sentences were added in the last paragraph of the Introduction section to emphasize the goal of this study:

The current study aimed to identify the points that should be considered when maintaining sufficient human resources to achieve the final goal of facilitating nuclear disaster response activities. To attain the abovementioned goals, the negative chain of stress and anxiety (Fig 1) should be inhibited, and this study aimed to provide a solution to this problem by assessing objective stress values.

Methodology: The study design should be briefly explained and justified with a brief sub-heading

Some subheadings (e.g., 2.1–2.5) were revised.

Discussion: The limitations of the study can be highlighted more clearly and its suggested that they are re-written.

Thank you for pointing this out. We have added a separate Limitation section. Moreover, we have declared that this study is observation during just a seminar, not the real radiation emergency situation as a study limitation. Further, it is unclear the kind of factor affects the stress during the decontamination process. Nevertheless, to determine specific strategies that should be used to reduce stress effectively, further studies should be conducted to identify factors associated with elevated stress levels.

Reviewer #5: Comments to the Author

I want to congratulate the authors on this clear and well-written manuscript.

Overall, this high-quality manuscript has implications for policymakers and other concerned bodies. But there are some points of consideration for revision, but these changes are not likely to be substantial. I, therefore, recommend the following minor revisions.

INTRODUCTION

- Well-written

- author state in line with 46 “In recent years” this is not a good word because what does it mean how many years are called as recent years, so please be specific or try to put the time period.

Thank you for pointing this out. The term was changed to “After the 1990s.”

- the author needs to revise the sentence In line with “57- 63” the statement was to wordy please try to summarize it.

The following sentences were edited:

“The nuclear disaster caused by an accident at the Fukushima Daiichi Nuclear Power Plant (FDNPP) after the Great East Japan Earthquake on March 11, 2011, is one of the most disastrous events in recent years. Crisis response professionals have concerns about the risk of radiation exposure in the general population after the FDNPP accident. During the initial response, hospital staff at medical institutions receiving victims experienced strong feelings of anxiety due to the risk of radiation exposure, which affected the smooth acceptance of patients in the hospital.”

- the sentence in line with “80-84” have to written here in the introductory part, I think they are part of result of this study.

According to your advice, the following sentences were revised to make them more understandable. Moreover, the legend of Figure 1 was retained.

“Therefore, the shortage of human resources in this field is attributed to the negative chain of events caused by anxiety and stress (Fig 1). Lack of knowledge causes radiation exposure-related anxiety among workers, which reduces their intention to engage (14). This resulted in a shortage of human resources and educators, thereby making it impossible to provide sufficient education and increase the knowledge of workers (14, 20-23). Hence, this negative chain should be suppressed to smoothly provide nuclear disaster medical services.”

METHODS

- well-written

- there is a need of separate subheading which indicates study area, study design, participant selection, and data collection technique.

We have revised the subheadings according to your advice. The Methods section was also modified. That is, “study design,” “patient selection,” and “data correction” subheadings were added to clarify the contents. Our study field was medical education.

RESULT

- well-written

- How authors classified that the level of Anxiety as “Extreme, Moderate, Little, and None” this not clear, because there have to be predefined criteria or scale of measure to classify the anxiety level.

The survey used a self-administered questionnaire with a 4-point Likert scale. The “2.2.2. Questionnaire survey on the background characteristics of the participants” section and Table 1 and 4 footnotes were added.

DISCUSSION

- Well-written

- LIMITATION OF THE STUDY: if possible, to mention it’s recommended.

Thank you for the advice. A separate Limitation section has been added.

- CONCLUSION AND RECOMMENDATION: - it’s good if there is conclusion and recommendation headings. Also, recommendation for future direction.

Thank you for the suggestion. A separate Conclusion section has been added.

---

## [Decision Letter · Decision Letter 1]

8 Aug 2022

PONE-D-22-11048R1Objective stress values during radiation emergency medicine for future human resourcesPLOS ONE

Dear Dr. Keita,

Thank you for submitting your manuscript to PLOS ONE. After careful consideration, we feel that it has merit but does not fully meet PLOS ONE’s publication criteria as it currently stands. Therefore, we invite you to submit a revised version of the manuscript that addresses the points raised during the review process.

We look forward to receiving your revised manuscript.

Kind regards,

Mohamad Syazwan Mohd Sanusi

Academic Editor

PLOS ONE

Additional Editor Comments :

Academic Editor's comments (Syazwan)

Title – human resource is too general. Can you be more specific which class of worker? Nurses?

Abstract – organisation OK

First para (Line 48-58, Page 46 PDF) – The introduction of the first paragraph is insignificant to this study. It is too general. Please rewrite and be more specific to the title.

First para - all is about manpower, human resource. The author should narrow it. Nurses? Can you define in first para which class of worker? Who is going to engage in radiation emergency medicine in case of an actual nuclear disaster?

First para – does the nurses participated in this study have working in nuclear disaster emergency response in Fukushima? If not the the title or anything related to nuclear disaster emergency response must be stripped, and replaced with clinical procedure/diagnostic/treatment using radiation/radionuclides or just nuclear medicine practices.

Line 96-98 – Redundant to a line in previous para.

Line 98-100 - Redundant to previous para

Line 104-105 – 20 seminar each year?

Line 119-124 – redundant

Line 127-129 – redundant

Line 137 – remove “As mentioned in the Introduction”

Data collection 2.3, Line 158 - Use of electrocardiogram (HRV, RR) to measure average stress values. This is the main instrument used for this work. I would like you to extend the discussion, reference of other studies that support this method in your “literature work part” in introduction. I believe it will be impactful to reader as readers need to know on what hyphothesis or basis that you used to measure stress value.

Line 192 – survey on background characteristics? Please clarify the justification of the need for this survey. Why it is not a survey to assess worker understanding on radiation exposure, effects and radiation emergency situation and risk? And the question is too brief to identify or confirm the alct of knowledge or fundamental among radiation emergency workers?

Methodology, Line 204 - Please extend the statistical technique used in this work? In order for the readers to understand the test and results, I believe it is useful to address the flowchart to describe what test has been used? And to test what parameter? Eg. T-test, z-test, ANOVA to check difference of background factors in two groups.

Line 256 – 260 - The highlighted text are appropriate in introduction.

Line 206 – 201 – remove. Please straight to the point. Discussion is to discuss your analysis/data obtained in Result

Line 288 -289 – “The present study found no difference in stress values between the group interested in nuclear disasters and that with no interest” need P-value that you have calculated to support.

Discussion – major revision. In discussion it would be useful to state you finding and cite your analysis values. Please revise thoroughly, each statement of finding given in discussion must come along with any measurable/evidence eg. P-values, % etc.

Conclusion – no conclusion has been drawn.

Reviewers' comments:

Reviewer's Responses to Questions

**Comments to the Author**

1. If the authors have adequately addressed your comments raised in a previous round of review and you feel that this manuscript is now acceptable for publication, you may indicate that here to bypass the “Comments to the Author” section, enter your conflict of interest statement in the “Confidential to Editor” section, and submit your "Accept" recommendation.

Reviewer #1: All comments have been addressed

Reviewer #3: All comments have been addressed

Reviewer #5: All comments have been addressed

2. Is the manuscript technically sound, and do the data support the conclusions?

Reviewer #1: Yes

Reviewer #3: Yes

Reviewer #5: Yes

3. Has the statistical analysis been performed appropriately and rigorously? 

Reviewer #1: Yes

Reviewer #3: Yes

Reviewer #5: Yes

4. Have the authors made all data underlying the findings in their manuscript fully available?

Reviewer #1: Yes

Reviewer #3: Yes

Reviewer #5: Yes

5. Is the manuscript presented in an intelligible fashion and written in standard English?

Reviewer #1: Yes

Reviewer #3: Yes

Reviewer #5: Yes

6. Review Comments to the Author

Reviewer #1: (No Response)

Reviewer #3: All corrections included. All methodology section of the article addressed. Understanding the fact that justifying the scientific process is mandatory for a sound and quality reasearch outcome and informed decision making .

All corrections included. All methodology section of the article addressed. Understanding the fact that justifying the scientific process is mandatory for a sound and quality reasearch outcome and informed decision making

All corrections included. All methodology section of the article addressed. Understanding the fact that justifying the scientific process is mandatory for a sound and quality reasearch outcome and informed decision making

Reviewer #5: (No Response)

7. PLOS authors have the option to publish the peer review history of their article (what does this mean?). If published, this will include your full peer review and any attached files.

Reviewer #1: No

Reviewer #3: No

Reviewer #5: **Yes: **Astawus Alemayehu

---

## [Author Response · Author response to Decision Letter 1]

19 Aug 2022

Title – human resource is too general. Can you be more specific which class of worker? Nurses?

Thank you for your comment. We have changed the title as follows: “Objective stress values during radiation emergency medicine for future human resources: Findings from a survey of nurses.”

Abstract – organisation OK

First para (Line 48-58, Page 46 PDF) – The introduction of the first paragraph is insignificant to this study. It is too general. Please rewrite and be more specific to the title.

Thank you for your advice. We have added the following sentences to show the significance of our study: “In particular, nurses account for approximately 40% of healthcare workers, and understanding their current status is crucial for efficiently securing human resources for the entire healthcare workforce.”

First para - all is about manpower, human resource. The author should narrow it. Nurses? Can you define in first para which class of worker? Who is going to engage in radiation emergency medicine in case of an actual nuclear disaster?

Thank you for your questions. The majority of healthcare workers are nurses, accounting for 40.2%, while physicians account for 10%. In practice, nurses also comprise the largest number of healthcare workers in radiation emergency medicine.

First para – does the nurses participated in this study have working in nuclear disaster emergency response in Fukushima? If not the the title or anything related to nuclear disaster emergency response must be stripped, and replaced with clinical procedure/diagnostic/treatment using radiation/radionuclides or just nuclear medicine practices.

Thank you for your comment. Since the nuclear disaster in Fukushima in 2011, Fukushima Medical University has accepted a large number of exposed and contaminated patients. We have responded to the nuclear disaster as an organization, and the nurses employed at our institution have earnestly acted. Thus, we believe that the title of our study is appropriate because the training given at our radiation emergency medicine seminars addresses the response to nuclear disasters.

We have changed the sentences in lines 90-93 as follows: “Since the FDNPP accident, Fukushima Medical University (FMU) has been working to accept patients exposed to or contaminated with radiation related to the FDNPP accident. Furthermore, FMU continues to develop human resources related to radiation emergency medicine (REM) and presents over 20 REM-related seminars annually.”

Line 96-98 – Redundant to a line in previous para.

Line 98-100 - Redundant to previous para

Thank you for noting these instances. We have removed the figure legends and retained only the figure title.

Line 104-105 – 20 seminar each year?

Yes, FMU conducts over 20 seminars annually. In other words, we hold seminars approximately twice a month.

Line 119-124 – redundant

We have deleted the text as per your comment.

Line 127-129 – redundant

We have deleted the text as per your comment.

Line 137 – remove “As mentioned in the Introduction”

We have deleted the text as per your comment.

Data collection 2.3, Line 158 - Use of electrocardiogram (HRV, RR) to measure average stress values. This is the main instrument used for this work. I would like you to extend the discussion, reference of other studies that support this method in your “literature work part” in introduction. I believe it will be impactful to reader as readers need to know on what hyphothesis or basis that you used to measure stress value.

Thank you for your valuable advice. We have added the following sentences to the Introduction section.

“A widely known method for objectively indicating stress levels is the evaluation of sympathetic nerve activity based on HRV (29, 30). HRV refers to the variation in the RR interval of the electrocardiogram, which indicates the instantaneous heart rate, and is affected by the balance between the sympathetic and parasympathetic nervous systems. The low-frequency/high-frequency ratio (LF/HF) obtained from the analysis of the HRV frequency is used as a surrogate parameter for sympathetic overactivity and is associated with stress load (31-33). There are several reports on the validity of HRV in assessing stress levels (34-37). Furthermore, with advances in technology, it has been suggested that wearable devices capable of measuring HRV can be used to determine stress levels (30, 38).”

 Line 192 – survey on background characteristics? Please clarify the justification of the need for this survey. Why it is not a survey to assess worker understanding on radiation exposure, effects and radiation emergency situation and risk? And the question is too brief to identify or confirm the alct of knowledge or fundamental among radiation emergency workers?

The primary endpoint of this survey was comparing the stress levels for each process during the REM seminar to identify those with the highest stress load. A difficult, complex, and time-consuming questionnaire may induce stress in itself and consequently affect the stress value of the seminar process. Therefore, the questions asking about background characteristics to be answered before and after the seminar were simply designed so that they could be completed within 5 minutes.

It has been reported that interest in emergency medicine and experience in disaster activities are related to an interest in nuclear disaster medicine, and the questionnaire in our study was developed with this in mind. In addition, we asked questions about the recognition and knowledge of radiation exposure and established their own acceptable standards of radiation exposure dose. Thus, the questionnaire responses represent the participants' understanding of radiation exposure, radiation effects, and radiation emergency situations.

We have added the following sentence to the revised manuscript: “Yamada et al. (41) reported that interest in emergency medicine and experience in disaster activities are related to an interest in nuclear disaster medicine, and the questionnaire in our study was developed with this in mind.”

Methodology, Line 204 - Please extend the statistical technique used in this work? In order for the readers to understand the test and results, I believe it is useful to address the flowchart to describe what test has been used? And to test what parameter? Eg. T-test, z-test, ANOVA to check difference of background factors in two groups.

As stated in section 2.4. (Statistical analysis), we used t test for age and Fisher's exact test for other background factors presented in Tables 1 and 4. Stress values were compared using t test and Tukey–Kramer test for multiple comparisons in Table 2, Table 3, and Fig 3.

Line 256 – 260 - The highlighted text are appropriate in introduction.

Thank you for pointing this out. We have moved these sentences from the Discussion to the Introduction section.

Line 206 – 201 – remove. Please straight to the point. Discussion is to discuss your analysis/data obtained in Result

We have responded to your comment assuming that you are referring to lines 266–271, not lines 206–201.

We interpreted your comment as a reference to the lack of result data values in the Discussion section of the manuscript. Thus, we have added results to the Discussion section of the revised manuscript.

Line 288 -289 – “The present study found no difference in stress values between the group interested in nuclear disasters and that with no interest” need P-value that you have calculated to support.

Thank you for your comment. We have added p-values to the revised manuscript.

Discussion – major revision. In discussion it would be useful to state you finding and cite your analysis values. Please revise thoroughly, each statement of finding given in discussion must come along with any measurable/evidence eg. P-values, % etc.

As per your comment, we have added p-values and numbers to the Discussion section of the revised manuscript.

Conclusion – no conclusion has been drawn.

Thank you for pointing this out. Accordingly, we have added the following sentence to the Conclusion section of the revised manuscript: “The use of a shirt-type electrocardiogram made it possible to assess stress levels during the activities presented at the REM seminar. The stress load was highest during the decontamination process.”

---

## [Editor Report · Decision Letter 2]

30 Aug 2022

Objective stress values during radiation emergency medicine for future human resources: Findings from a survey of nurses

PONE-D-22-11048R2

Dear Dr. Ayama,

We’re pleased to inform you that your manuscript has been judged scientifically suitable for publication and will be formally accepted for publication once it meets all outstanding technical requirements.

Kind regards,

Mohamad Syazwan Mohd Sanusi

Academic Editor

PLOS ONE

---

## [Editor Report · Acceptance letter]

1 Sep 2022

PONE-D-22-11048R2 

Objective stress values during radiation emergency medicine for future human resources: Findings from a survey of nurses 

Dear Dr. Iyama:

I'm pleased to inform you that your manuscript has been deemed suitable for publication in PLOS ONE. Congratulations! Your manuscript is now with our production department. 

Kind regards, 

on behalf of

Dr. Mohamad Syazwan Mohd Sanusi 

Academic Editor

PLOS ONE